# Effect of a Four-Week Soccer Training Program Using Stressful Constraints on Team Resilience and Precompetitive Anxiety

**DOI:** 10.3390/ijerph20021620

**Published:** 2023-01-16

**Authors:** Juan Martin Tassi, Jesús Díaz-García, Miguel Ángel López-Gajardo, Ana Rubio-Morales, Tomás García-Calvo

**Affiliations:** Faculty of Sport Sciences, University of Extremadura, 10003 Cáceres, Spain

**Keywords:** cognitive exertion, stress, internal load, mental demands, soccer constraints, football, team sports

## Abstract

The present study examined the effects of stressful constraints during soccer trainings on psychological skill development and internal load when compared with control (nonstressful) trainings. A total of 51 elite male youth soccer players (27 in the experimental group, M = 16.54 years; 24 in the control group, M = 15.44 years) participated in the study. In a 12-week longitudinal survey, team resilience, using the Spanish version of the Characteristics of Resilience in Sports Teams Inventory, and anxiety, using the Sport Anxiety Scale, were measured at baseline (after 4 weeks of regular trainings), postprotocol (after 4 weeks of control or experimental trainings), and follow-up (after 4 weeks of regular trainings). Results show that, when compared with the control group, a program with stressful constraints helped young soccer players to develop better psychological skills: specifically, increased ability to cope with impairments in resilience (both resilience characteristics and team vulnerability under pressure; *p* < 0.001). Increases in anxiety (*p* = 0.06) and decreases in preoccupation (*p* < 0.001) and lack of concentration (*p* < 0.001) were also observed. The adaptation of human behavior to specific trainings may explain these results. In conclusion, the regular exposure of young soccer players to stressful situations during trainings shows benefits for their psychological skill development in soccer. Then, benefits on internal load were also observed.

## 1. Introduction

Soccer is a physically as well as mentally demanding activity [1]. The interaction of a soccer player with their sport environment leads to many factors such as stress, cognitive load, or concentration that contribute to sport outcomes [2]. Despite the notable advances in technology or statistics to the quantification of physical efforts in soccer training and game scenarios [3], human factors continue to influence the outcomes [4]. As for physical fatigue, where soccer coaches may increase or decrease physical efforts to induce adaptations or achieve better performance during competitions, it is necessary to control the psychological load and adaptations of psychological skills (derived from human factors) [1]. To our knowledge, this is the first study that examines the effects of a training program adaptation focused on the effect of stressful situations on soccer-specific psychological skill development.

Soccer is primarily a low-intensity endurance activity interspersed with repeated short bouts of high-intensity activity [1]. However, soccer is also cognitively demanding and fatiguing [5,6]. It has been stated that soccer is one of the highest cognitive demand sports due to its performance being influenced by high levels of attention and frequent decision making [7]. Soccer players’ capacity to tolerate the negative effects of mental exertion on performance [8,9,10] or to maintain attention and cognitive processing during stressful, high time pressure or adverse soccer scenarios [11,12] determines their outcomes and physical or technical–tactical performance. During a soccer season, sports teams experience adverse situations that can increase players’ stress [13]. Stress has been defined as “a psychological condition that occurs when people observe a substantial imbalance between the demands they endure and their ability to meet them, and when that inability has significant consequences” [14]. During competitions, these stressful events can impair their technical–tactical actions or produce recurrent states of anxiety. Therefore, it is not surprising that coaches seek strategies to counter stress and its effects on performance.

The conditional demands of soccer and their trainability have been widely studied. Coaches are encouraged to design the main physical capacities during the trainings carried out from Match Day (MD)-4 to MD-2 (i.e., 4 and 2 days before Match Day, respectively) and to avoid physical fatigue shortly before the next match (i.e., MD-1 and MD) in typical microcycles [15]. For this purpose, the effects of different constraints on physical efforts and fatigue have been examined [16,17,18,19]. The authors concluded that, in general, constraints that allow players more space (e.g., a higher ratio of players per meter) and more time playing (e.g., fewer technical–tactical difficulties) increased physical demands. Thus, coaches can decide whether to use these constraints according to their effects on physical fatigue and the coaches’ objectives.

The effects of constraints on psychological demands have also been examined, although to a lesser extent. It has been shown that an adverse match outcome [11], higher time pressure (i.e., less time to make decisions) [20], the coach’s active participation [21], or an increase in the task’s technical-tactical difficulty [22] significantly increase the task’s stress and psychological demands. However, the effects of these constraints on an intervention program and the development of psychological skills remain unknown. Nonetheless, this information may help coaches to decide whether to use these adaptations according to their objectives and the week’s training. Ferreira et al. [23] suggested that training under stressful conditions may lead players to create specific mental adaptations to minimize the effects of stress on performance. Planning for stressful stimuli perceived as threats during specific training tasks would increase players’ resilient characteristics [24] and decrease their anxiety levels [25,26]. Morgan et al. [27] defined team resilience as “a dynamic, psychosocial process which protects a group of individuals from the potential negative effect of stressors they collectively encounter.” Thus, it is important to link tactical contents in different team sports [28,29,30,31,32] to pressure and scenarios of coping with stressful situations to determine how they influence team resilience and players’ anxiety [33,34].

### The Present Study

Given the lack of intervention programs that associate tactical and psychological aspects related to stressful scenarios, the main objective of the study was to analyze the effects of an intervention employing stressful and adverse constraints on the development of tactical components in soccer training tasks. For this purpose, a systematic plan was developed that integrated different stressful constraints through tasks and game situations. Based on previous evidence, the main hypothesis of the study states that the intervention program is expected to improve team resilience (i.e., increase players’ resilient characteristics and decrease vulnerability under pressure in the team) and decrease anxiety levels.

Through the present intervention, it will be possible to differentiate the assessment and interventions on individual and team aspects [35], which will improve the specificity and individualization of intervention plans in future training tasks. Coaches can direct the training tasks through regular game actions in which the rules can be modified, or the constraints favoring this type of adverse scenarios can be established [32].

Regarding the duration of the intervention program, we took as reference prior works oriented to training situations under pressure [36,37] or manipulating environmental or task demands [32,33,34,35,36,37,38]. However, no works show the importance of a specific time to develop or improve psychological or mental capacities related to performance through tactical tasks. This research would produce information about the characteristics and duration of the type of task required to develop tactical and psychological variables and aspects in an integrated way. This plan would justify future interventions using conditions of tasks under pressure such as game actions and would be useful to improve team resilience and reduce players’ anxiety [35]. Therefore, this proposal deepens soccer training methodologies and promotes interdisciplinary work that considers specific training tasks with tactical components as a substantial factor.

Given the importance of psychological factors for soccer performance and the lack of previous studies on how to develop psychological skills through specific soccer trainings using constraints, this study will attempt to examine the effect of an intervention program using stressful constraints on the development of soccer-specific psychological skills (i.e., team resilience and precompetitive anxiety). 

## 2. Materials and Methods

### 2.1. Participants

A total of 51 elite male youth soccer players from the academy of an elite Argentinean team participated in the study. Of them, 27 players of the under-17-year team in the academy comprised the experimental group (M =16.54 ± 1.23years), and 24 players of the under-16-year team made up the control group (M =15.44 ± 1.09years). More information about players is included in Table 1.

### 2.2. Instruments

#### 2.2.1. Team Resilience

Team resilience was measured with the Spanish version of the Characteristics of Resilience in Sports Teams Inventory (CREST; [39]), validated by López-Gajardo et al. [40]. This questionnaire begins with a stem phrase (e.g., “In the last month, when my team was under pressure...”) followed by a total of 20 items divided into two factors: Characteristics of Resilience (12 items, e.g., “the team gained confidence by working together to overcome pressure”) and Vulnerability under Pressure (8 items, e.g., “the team couldn’t resist at the most difficult times”). Responses are rated on a 9-point Likert scale ranging from 1 (totally disagree) to 9 (totally agree). The Confirmatory Factor Analysis with two factors showed an adequate fit of the model, χ2(169) = 358.089, *p* < 0.001, Comparative Fit Index = 0.934, Tucker–Lewis Index = 0.926, Root Mean Square Error of Approximation = 0.057, 95% CI [0.049, 0.065]), and Standardized Root Mean Square Residual = 0.043. Similarly, adequate values of internal consistency for Characteristics of Resilience (α = 0.93, ω = 0.88) and Vulnerability under Pressure were also obtained (α = 0.93, ω = 0.88). 

#### 2.2.2. Sport Anxiety Scale

The Sport Anxiety Scale (SAS-2) was designed to assess precompetitive anxiety in a sports context [41,42]. It includes 15 items divided into three factors: Somatic Anxiety (five items. e.g., “My body feels tense”), Worry (five items. e.g., “I worry that I will not play my best”), and Concentration Disruption (five items. e.g., “It is hard to concentrate”). All items were rated on a Likert scale ranging from 1 (Not at all) to 4 (A lot). The Confirmatory Factor Analysis showed an adequate fit to the data: χ2(48) = 113.692, *p* < 0.001, Comparative Fit Index = 0.940, Tucker–Lewis Index = 0.917, Root Mean Square Error of Approximation = 0.063 (95% CI [.048, 0.078]), and Standardized Root Mean Square Residual = 0.057. Regarding the internal consistency of this measure, we obtained the following Cronbach’s alpha and omega coefficients for Anxiety (α = 0.83; ω = 0.75), Worry (α = 0.83; ω = 0.75), and Concentration Disruption (α = 0.81; ω = 0.77).

### 2.3. Procedure

This intervention program was approved by the Ethics Committee of the University of the first author, following the Declaration of Helsinki AMM. The ethical requirements of the American Psychological Association (APA) were also met. The teams were contacted and informed about the stages and objectives of the investigation. Following their agreement, all the players were informed of the procedure to be followed, emphasizing their voluntary participation and the confidentiality of their responses. All the players included in the intervention gave their consent. 

A quasi-experimental study was performed using a repeated-measures design: preintervention, postintervention, and follow-up, with two independent groups (control group and an experimental group). The study was performed over a total of 12 weeks, divided into three phases. Firstly, from Week 1 to Week 4, we administered the questionnaires to establish a baseline (Phase 1: preintervention; PRE). Secondly, from Week 5 to Week 8, we conducted the intervention during four training sessions per week (Phase 2: postintervention; POST). We introduced constraints in the soccer tasks of the experimental group to increase the mental load and stress of these soccer tasks. The main objective of this design was to test the effects of increasing the psychological demands on the above-mentioned psychological variables (team resilience and precompetitive anxiety). This program was developed by 10 expert university professors with experience as soccer coaches and focused on soccer and mental demands. Examples of tasks implemented in the experimental group are: one team started to lose, or one team had limited time to achieve its goal. These constraints were not implemented in the control group. Table 2 shows more examples of the tasks designed for the experimental group. Thirdly, from Week 9 to Week 12, we administered the questionnaires again to determine the effect of the intervention (Phase 3: follow-up; SEG). 

### 2.4. Data Analysis

Data were analyzed using the statistical package for social sciences SPSS 25.0 (2017) (IBM, Armonk, NY, USA). Data are presented as mean ± standard deviation. Firstly, the normality of the data was confirmed by the Shapiro–Wilk test. Then, parametric tests were used. One-way analysis of variance (ANOVA) was used to test group differences in the dependent variables (i.e., team resilience and precompetitive anxiety) before starting the study. When significant group differences were observed in premeasures, these measures were included as covariates. To determine the effect of the intervention program, we used a 2-way ANOVA with time and condition as factors (3 Times (Preintervention, Postintervention, and Residual) × 2 Conditions (Experimental and Control)). If this analysis yielded a significant interaction effect, the main effect of time (Pre, Post, Seg) was calculated for each condition. Post hoc analysis using Tukey was performed to also verify the analysis performed. Significance was set at *p* < 0.05.

## 3. Results

Table 3 shows the results of team resilience and precompetitive anxiety in both groups for PRE, POST, and SEG. Before beginning the intervention, there were no significant group differences in Resilience (Characteristic) (*p* = 0.78), or Concentration Disruption (*p* = 0.27). However, there were significant group differences in Resilience (Vulnerability) (*p* = 0.02), Anxiety (*p* = 0.02), and Worry (*p* < 0.001).

A significant Condition × Time interaction was observed for Resilience (Characteristics) (*p* < 0.001). The main effect of time revealed no significant changes in the experimental group in Resilience (Characteristics) (*p* = 0.61), whereas a significant decrease was found in the control group (pairwise comparisons showed a significant decrease (*p* < 0.001) from PRE and POST to SEG (*p* < 0.001)). A significant Condition × Time interaction was observed for Resilience (Vulnerability) (*p* < 0.001). The main effect of time showed no significant changes in the experimental group in Resilience (Vulnerability) (*p* = 0.64) and a significant progressive increase (PRE < POST < SEG) in the control group (*p* < 0.001). A nearly significant Condition × Time interaction was observed for Anxiety (*p* = 0.06). The main effect of time showed no significant effect of time in Anxiety (*p* = 0.68 in both cases) in the experimental group, whereas a significant increase in Anxiety from PRE and POST to SEG (*p* < 0.001 in both cases) was observed in the control group. A significant Condition × Time interaction was observed for Worry (*p* < 0.001). The main effect of time and the pairwise comparisons showed a significant decrease in Worry from PRE to POST (*p* < 0.001), but followed by nonsignificant changes from POST to SEG (*p* < 0.41) in the experimental group. A significant decrease in Worry was found in the control group from PRE to POST (*p* < 0.001), followed by a significant increase from POST to SEG (*p* < 0.001). Then, the main difference between groups on this variable was observed in the change from POST to SEG, as observed in the Tukey Analysis. The experimental group also showed a significant Condition × Time interaction for Concentration Disruption (*p* < 0.001). The main effect of time showed a significant decrease in Concentration Disruption in the experimental group from PRE to POST (*p* < 0.001) and from PRE to SEG (*p* = 0.03), whereas a significant increase was shown in Concentration Disruption in the control group (*p* < 0.001).

## 4. Discussion

The main purpose of the present study was to determine the effects of stressful constraints during trainings on the development of psychological skills in young soccer players. The findings indicate that, compared with a control group, using stressful constraints helps soccer players decrease Worry and Concentration Disruption. Moreover, it does not impair their resilience (both as a characteristic and as vulnerability) or increase their anxiety. 

Most of the literature published in soccer about adaptations to training has focused on physical aspects [1]. This literature agrees that physical adaptations are determined by the specificity of the training. Although this has not been previously studied, Russell et al. [4] suggested that the training adaptations of psychological skills may be specific of training inputs. Therefore, and based on the results of the present study, strategies aimed at tactical development (principles of game linked to sport aggressiveness) and psychological development (resilience, sacrifice, attention, concentration, and anxiety related to stressful situations and coping with adverse situations) through specific soccer training tasks would favor the development of such psychological variables [13,36] (i.e., improving resilience characteristics and vulnerability under pressure, and decreasing anxiety, worry and concentration disruption). To our knowledge, this is the first study that confirms that using stressful constraints improves the development of psychological skills in young soccer players. Thus, training scenarios designed to provoke stress, adversity, and coping with adverse situations through aggressive sports actions is a relevant methodological aspect in tasks and situations similar to soccer competitions. This would reinforce the systematicity and integration of the performance components in soccer, especially focusing on the most ignored component (i.e., psychological skills). It would also enhance the development of skills associated with the psychological components [38], favoring the use of resources and tools in young soccer players’ training stage and improving professional soccer players. It has been shown that psychological skills may determine soccer performance [1], so such skills should be well developed. 

The analysis of the decline of the factors and capacities in the experimental group at postintervention may indicate a partial loss of capacities or a scenario of pressure and stress that exceeds their possibilities of adaptation [43]. On the other hand, they may achieve their best form in the following periods [44], as occurs with conditional capacities (e.g., repeated sprint ability, endurance). This is caused by the adaptation that players perform to the load of the training. The acute load caused by training may cause an acute reduction in the physical or physiological capacities, which are followed up by a chronic improvement on this capacity (adaptation). This has been highlighted with the implementation of the Brain Endurance Training, which is based on the development of physical and cognitive skills at the same time [45]. The experimental condition maintained the acquired psychological capacities to a greater extent. The results of the study confirm a better development of these capacities and also indicate their better maintenance over time. This is very important to achieve a good performance throughout an entire season and for the development of young soccer players and their future as professional soccer players.

Finally, this approach reveals the importance of coaches’ and physical trainers’ knowledge of psychological and mental variables and methodological processes to integrate them into the training and development of other capacities. At the same time, it highlights the contribution of psychologists and interdisciplinary work within specific training practice with strategies that incorporate and relate the technical–tactical contents of planning.

## 5. Limitations

Although the authors think that the findings of the present study are of interest, the study has several limitations, and the findings should be addressed carefully. The first limitation is the sample. We would like to highlight the difficulty to perform this type of ecological study; indeed, the sample is formed by youth elite players. However, future studies should replicate the findings in different contexts and large samples. The second limitation is the variables used. We would like to highlight the novelty of the variables used, their importance on performance, and the quality of the design (two groups × three times). However, future studies should replicate the findings of the present study but also include other related variables. Indeed, all the variables used were subjective perceptions, and future studies may use behavioral (e.g., reaction time) or physiological (e.g., heart rate variability) indicators of the brain related.

## 6. Conclusions

The results of the present study suggest that a 4-week program of soccer trainings using constraints to increase the mental stress of the tasks has positive effects on the development of psychological skills when compared with regular control trainings. Specifically, results show that, compared with a control group, the program helps soccer players decrease their worry and concentration disruption. Furthermore, it does not impair resilience (either characteristic or vulnerability) or increase precompetitive anxiety. Indeed, the follow-up measure performed four weeks after the program ended suggests that these positive psychological skills are maintained for a long time compared with the control group.

## Figures and Tables

**Table 1 ijerph-20-01620-t001:** Characteristics of the participants.

Variables	Experimental Group	Control Group
Age (years)	16.54 ± 1.23	15.44 ± 1.09
Height (cm)	175.09 ± 7.16	174.78 ± 9.67
Weight (Kg)	62.44 ± 10.31	61.16 ± 13.66
Experience in soccer (years)	9.09 ± 2.31	8.39 ± 1.16
Years in the club (years)	4.37 ± 1.12	3.99 ± 1.06

Notes. Values are presented as mean ± standard deviation.

**Table 2 ijerph-20-01620-t002:** Overview of the program intervention.

Variables	Strategies	Constraints Example
Resilience	Soccer players were exposed to increased mental demands (consequences) resulting from environmental and task demands. Specifically, regular exposure to pressure and demanding situations during soccer training tasks. Developing tasks were with negative consequences for noncompliance with instructions.	The coach removes a player from the team who does not respect the given instruction for one minute. The coach takes possession away from teams that lose the ball twice in a row.
Anxiety	Task planning under stressful situations, with tactical objectives or problems to be solved in a given space and time.	The coach gives the team 15 s of ball possession to take the ball from a defensive sector of the field towards the opponent’s goal and finish with crosses or shots.
Worry	Developing activities in which players worry about achieving and accomplishing more than one objective in the same task.	The coach sets two objectives in the same task to convert a point or goal: one will be in defense, and the other will be in attack (keeping a clean sheet for five minutes will allow them to perform combined finishing actions without opposition in the goal).
Concentration Disruption	Strategies are proposed in which the players must link and complete more than one general objective within the same task without forgetting any of them.	The coach places three balls in play during a 2 vs. 2 possession + 2 jokers (three in total) and must hold them for 5 s each. Then, the coach indicates which two players will defend and recover a ball in a maximum of 8 s.

**Table 3 ijerph-20-01620-t003:** Changes in psychological variables during 12 weeks. A group comparison.

Variables	Control Group			Experimental Group			Time × Condition
	Pre	Post	Seg	Pre	Post	Seg	
Resilience (Characteristic)	5.80 ± 0.58	5.40 ± 0.69	4.95 ± 0.53	5.73 ± 0.69	5.42 ± 0.39	5.69 ± 0.63	*F* = 13.42; *p* = *
Resilience (Vulnerability)	1.93 ± 0.43	2.80 ± 0.49	2.75 ± 0.80	2.51 ± 0.98	2.62 ± 0.57	2.32 ± 0.90	*F* = 9.58; *p* = *
Anxiety	8.40 ± 2.87	8.36 ± 2.37	9.64 ± 2.34	7.62 ± 1.82	7.06 ± 2.71	7.50 ± 2.00	*F* = 3.00; *p* = 0.058
Worry	14.48 ± 3.00	11.48 ± 2.21	12.96 ± 2.38	12.00 ± 2.73	10.25 ± 3.51	10.87 ± 3.34	*F* = 14.08; *p* = *
Concentration Disruption	7.64 ± 2.11	8.44 ± 1.85	13.08 ± 5.71	8.12 ± 1.20	7.25 ± 1.80	7.87 ± 2.33	*F* = 13.04; *p* = *

Notes. Values are presented as mean ± standard deviation. * *p* < 0.05.

## Data Availability

The data are not available due to the Football Club’s policy.

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
