# Peer review of "Effect of a Four-Week Soccer Training Program Using Stressful Constraints on Team Resilience and Precompetitive Anxiety"

_ijerph, 2023, doi:10.3390/ijerph20021620_

Round 1

Reviewer 1 Report

1. Throughout the text, the word MAGE is used. I don't understand what you mean by MAGE.

2. You must put the results summarized in the abstract

3. You must make a summary image of the abstract and place it in "supplementary files"

4. Introduction. I think you should do a slight review of English.

5. SD= replace with a sign of ("plus or minus") as shown in the table

6. In point 2.2 you must put what the different acronyms mean (e.g. CFI, TLI, RMSEA, etc etc)

7. H-CFA what is it?

8. You describe several variables as follows ( =.83; =.75) nothing is missing before the = sign ?

9. Table 2 - There are 2* in the caption with different meanings

10. must make a table with the characteristics of the participants in baseline 

11. Ref 45 is from 1950 (almost 75 years old)- DOESN'T THERE ANY MORE RECENT?

Author Response

Thanks for the opportunity to send a new version of the manuscript. Thanks for the time and improvements to our work.

Reviewer 2 Report

Manuscript ID: ijerph-2099003-peer-review-v1

Journal: International Journal of Environmental Research and Public Health

Effect of a 4-week soccer training program using stressful constraints on team resilience and pre-competitive anxiety

Thank you for the opportunity to review this manuscript. This is an interesting and novel topic adopting a novel scientific approach. The rationale is clearly presented and the methodology is generally well presented. I wonder whether some discussion is needed on individual resilience versus team resilience and their relationships to performance. You also offer no commentary on the age of the participants. I would have thought 15 and 16 year olds might present different challenges regarding resilience, particularly with the absence of senior leadership within a team.

Data analysis

L 173 - do you mean - mean ± standard deviation - I am not sure what medium means?

L178-179

Do you mean a 2 way ANOVA with condition and time as your two factors?

L181 - 182

You need to select one level of significance - probably p< 0.05

Results

Table 2 needs correcting and improving

Across the top line you have 2 X Control groups - presume one is the intervention but I am not sure which

Your next line is also confusing - presume you would have pre - post - residual for each group - again I am not sure which data belongs where and I am not where SEG came from - please be consistent with labels.

You have reported interactions which simply tells us that the groups behaved differently over time - did you perform post hoc analyses to determine where those differences might lie (e.g Tukey)

Without being able to follow and understand your statistical analysis I could not really comment on your ensuing discussion. It did however strike me that Selye’s GAS is a bit of a long reach - you have read/referenced his work, so will know that this has been mis/overinterpreted by the exercise science community. 

Please consider including a section on limitations (as noted above) in your conclusions section. 

L 51

Can you find a better word than ‘worsen’?

L 54 

The state-of-art of conditional demands and trainings

L72

 ‘create adaptations’ suggests that this is deliberate - possibly they adapt 

Author Response

Thanks for the opportunity to send a new version of the manuscript. Thanks for the time and improvements to our work.

Reviewer 2

Thank you for the opportunity to review this manuscript. This is an interesting and novel topic adopting a novel scientific approach. The rationale is clearly presented and the methodology is generally well presented. I wonder whether some discussion is needed on individual resilience versus team resilience and their relationships to performance. You also offer no commentary on the age of the participants. I would have thought 15 and 16 year olds might present different challenges regarding resilience, particularly with the absence of senior leadership within a team.

            General response: Thanks to Reviewer 2 for take his/her time to improve our manuscript. Thanks for your positive comment to our work as well. We have attended all your comments, which has been marked in red color to facilitate your new revision. A native English speaker has reviewed the manuscript according to your suggestion indeed.

Data analysis

L 173 - do you mean - mean ± standard deviation - I am not sure what medium means?

            Response: Sorry for this mistake. We changed it by “mean”. Thanks.

L178-179

Do you mean a 2 way ANOVA with condition and time as your two factors?

             Response: Yes. We have clarified it.

L181 - 182

You need to select one level of significance - probably p< 0.05

            Response: Ok. It was changed in Data Analysis and Table 2.

Results

Table 2 needs correcting and improving

Across the top line you have 2 X Control groups - presume one is the intervention but I am not sure which

            Response: Sorry for the mistake. We indicated two control groups in the previous version. We have indicated which is the data of the Experimental Group. Sorry.

Your next line is also confusing - presume you would have pre - post - residual for each group - again I am not sure which data belongs where and I am not where SEG came from - please be consistent with labels.

            Response: Sorry, this was a mistake. It has been changed accordingly.

You have reported interactions which simply tells us that the groups behaved differently over time - did you perform post hoc analyses to determine where those differences might lie (e.g Tukey)

            Response: Thanks to the reviewer for this comment. The 2-way ANOVA showed us that the evolution of the values in time was different between groups in the case of a significant interaction condition x time. Then, the analysis of time allows us to identify where these differences were. Based on your comment and due to Tukey and ANOVA are complementary, we have repeated the analysis using Tukey as indicated in the Data Analysis and we have checked again where these differences are. The results are the same, but we think that this new analysis is more strength. We have tried to improve the description of the results in the variables where the results may be confused based on Tukey indeed. Thanks again.

Without being able to follow and understand your statistical analysis I could not really comment on your ensuing discussion. It did however strike me that Selye’s GAS is a bit of a long reach - you have read/referenced his work, so will know that this has been mis/overinterpreted by the exercise science community.

            Response: Thanks for this interesting comment. We have deleted this reference. Indeed, we have focused this part on the adaptations of Brain to training demands.

Please consider including a section on limitations (as noted above) in your conclusions section.

            Response: Thanks! We think that it is so important for our work. We have included it.

L 51 Can you find a better word than ‘worsen’?

            Response: We have changed “worsen” by “impair”.

L 54 The state-of-art of conditional demands and trainings

            Response: It has been changed.

L72 ‘create adaptations’ suggests that this is deliberate - possibly they adapt

            Response: This study suggest that coaches may adapt the mental load and the stress of the training situations. Moreover, the authors indicated that the mental demands of the training can produce specific adaptations, as occurred with physical capacities. We have clarified it. Indeed, the rest of the references are in line with this affirmation. Thanks to the reviewer for this indication.

Round 2

Reviewer 1 Report

3. You must make a summary image of the abstract and place it in "supplementary files"

Author Response

Thanks again for your time to improve our work.

We have provided the summary image of the abstract.

Reviewer 1

            General response: Thanks to Reviewer 1 for take his/her time to improve our manuscript. We have attended all your comments, which has been marked in red color to facilitate your new revision. A native English speaker has reviewed the manuscript according to your suggestions.

  1. Throughout the text, the word MAGE is used. I don't understand what you mean by MAGE.

            Response: This is the abbreviation of medium age, and “y” is the abbreviation of years. Sorry for this mistake. We have changed: “Mage =15.44y” by  “M = 15.44 years” according to the rules of the journal.

  1. You must put the results summarized in the abstract

            Response: Thanks for the comment. We have included specific information of results in the abstract.

  1. You must make a summary image of the abstract and place it in "supplementary files"

            Response: Done.

  1. Introduction. I think you should do a slight review of English.

            Response: It has been performed by a native speaker. Thanks!

  1. SD= replace with a sign of ("plus or minus") as shown in the table

            Response: Thanks. It was changed.

  1. In point 2.2 you must put what the different acronyms mean (e.g. CFI, TLI, RMSEA, etc etc)

            Response: Because we only used two times all of these, the complete name of these index was included: “Comparative Fit Index”, “Tucker-Lewis Index”, “Root Mean Square Error of Approximation”, “Standardized Root Mean Square Residual”.

  1. H-CFA what is it?

            Response: Sorry it was a mistake. We included the complete name of the index: Confirmatory Factor Analysis.

  1. You describe several variables as follows ( =.83; =.75) nothing is missing before the = sign ?

            Response: Sorry for the mistake. We have included the correct symbols: “(α = …; ω = …)”.

  1. Table 2 - There are 2* in the caption with different meanings

            Response: Sorry for this mistake. It was changed.

  1. must make a table with the characteristics of the participants in baseline

            Response: It has been included as Table 1.

  1. Ref 45 is from 1950 (almost 75 years old)- DOESN'T THERE ANY MORE RECENT?

            Response: It was changed. Thanks!